# Photolysis and TiO$_2$ Photocatalytic Treatment under UVC/VUV Irradiation for Simultaneous Degradation of Pesticides and Microorganisms

**Sung Won Choi** [1] ⓘ**, Hafiz Muhammad Shahbaz** [2] ⓘ**, Jeong Un Kim** [1]**, Da-Hyun Kim** [1]**, Sohee Yoon** [1]**, Se Ho Jeong** [3]**, Jiyong Park** [1,*] ⓘ **and Dong-Un Lee** [3,*] ⓘ

[1] Department of Biotechnology, Yonsei University, Seoul 03722, Korea; wonchoice@yonsei.ac.kr (S.W.C.); gochief@yonsei.ac.kr (J.U.K.); apple7172@yonsei.ac.kr (D.-H.K.); sunnyoon90@yonseidairy.com (S.Y.)

[2] Department of Food Science and Human Nutrition, University of Veterinary and Animal Sciences, Lahore 54000, Pakistan; muhammad.shahbaz@uvas.edu.pk

[3] Department of Food Science and Technology, Chung-Ang University, Anseong 17546, Korea; mrr003@cau.ac.kr

* Correspondence: foodpro@yonsei.ac.kr (J.P.); dong-un.lee@cau.ac.kr (D.-U.L.)

**Abstract:** Efficiencies of various treatments for UVC photolysis (ultraviolet light-C at 254 nm), VUV photolysis (vacuum ultraviolet light at 254 nm and 185 nm), UVC-assisted titanium dioxide photocatalysis (UVC-TiO$_2$), and VUV-assisted titanium dioxide photocatalysis (VUV-TiO$_2$) were investigated for the degradation of pesticides including pyraclostrobin, boscalid, fludioxonil, and azoxystrobin and inactivation of microorganisms *Escherichia coli* K12 as a surrogate for *E. coli* O157:H7 and *Saccharomyces cerevisiae* in aqueous solutions and on the surface of fresh cut carrots. The degradation efficiencies of VUV were higher than for UVC on pesticides in aqueous solutions. However, there was no significant difference between degradation efficiencies for UVC and UVC-TiO$_2$ treatments, and between VUV and VUV-TiO$_2$ treatments. UVC, VUV, UVC-TiO$_2$, and VUV-TiO$_2$ showed similar inactivation effects against *E. coli* K12 and *S. cerevisiae* in aqueous solutions. The combined use of UVC and VUV treatments (combined UV) and combined use of UVC-TiO$_2$ and VUV-TiO$_2$ treatments (combined UV-TiO$_2$) showed higher efficiencies (72–94% removal) for the removal of residual pesticides on fresh cut carrots than bubble water washing (53–73% removal). However, there was no significant difference in removal efficiency between combined UV and combined UV-TiO$_2$ treatments. For *E. coli* K12 and *S. cerevisiae* on fresh cut carrots, the combined UV-TiO$_2$ treatment (1.5 log and 1.6 log reduction, respectively) showed slightly higher inactivation effects than combined UV (1.3 log and 1.2 log reduction, respectively). Photolysis and TiO$_2$ photocatalytic treatments under UV irradiation, including VUV as a light source, showed potential for the simultaneous degradation of pesticides and microorganisms as a non-chemical and residue-free technique for surface disinfection of fresh produce.

**Keywords:** photolysis; UVC; VUV; TiO$_2$ photocatalysis; pesticide; microorganism; fresh produce

## 1. Introduction

An increased consumption of fresh fruits and vegetables has led to an increased risk of foodborne illnesses that are associated with pathogens and toxic chemical residues [1,2]. Pesticides are representative of chemical hazards and their use has increased for controlling diseases and pest insect infestation for increasing crop yields. However, pesticides may have toxic effects on humans who consume fresh produce [3–7]. Furthermore, pesticides do not always stay in the location where they were applied and move along in the environment and often contaminate water, air, and soil, even in



remote areas. The toxicities of pesticides to organisms, including beneficial insects and non-target plants, can change the natural balance of the ecosystem by altering the environment to favor the pests [8]. In this study, four pesticides (pyraclostrobin, boscalid, fludioxonil, and azoxystrobin) that are widely used during the growth of various vegetables and fruits were chosen for the degradation experiment [9–12].

Pathogens, which are biological hazards, can easily contaminate fresh produce at any point in the food supply chain, leading to foodborne illness, outbreaks, and even death [1,13]. There is a need for effective and non-toxic post-harvest decontamination interventions for the degradation of pesticide residues and microorganisms. Photolysis is one of the most important degradation mechanisms for pollutants such as pesticides. Photolytic methods are based on providing the high energy to chemical compounds in the form of photon [14]. Most pesticides show UV–Vis absorption bands at relatively short UV wavelengths. UV irradiation leads to the promotion of the pesticides to their excited singlet states, and such excited states can then undergo homolysis, heterolysis, and/or photoionization [15]. However, sunlight reaching the earth's surface (mainly UVA, with varying amounts of UVB) only contains a very small amount of short wavelength UV radiation; therefore, the direct photolysis of pesticides by sunlight is generally expected to be limited. Thus, direct UV irradiation at short wavelength causes characteristic reaction such as bond scission, cyclization, and rearrangement, which are scarcely observed in hydrolysis and microbial degradation [14]. The intensity and wavelength of the UV radiation or the quantum yield of the compound to be eliminated are factors that affect the performance of the photodegradation process. As a source of UV radiation, mercury vapor lamps are usually used [16].

Advanced oxidation processes (AOPs) have also been reported as effective post-harvest decontamination methods. AOPs refer to oxidation processes that involve the generation of hydroxyl radicals (OH·) and strong oxidants at sufficient concentrations. Hydroxyl radicals (OH·) can be produced through the applications of oxidizing agents (such as $H_2O_2$ and $O_3$), UV irradiation, ultrasound (also called microbubble), electric current, and catalysts (such as $Fe^{2+}$) [17]. Hydroxyl radicals (OH·) are reactive oxidizing compounds with an oxidation potential of 2.80 eV, which is more reactive than both ozone (2.08 eV) and chlorine dioxide (1.36 eV). Furthermore, hydroxyl radicals (OH·) have the advantage of no additional waste generation, no chemical toxicity, and no corrosiveness to equipment. However, the disadvantage of AOPs lies in their high cost due to the use of expensive reagents ($H_2O_2$, for example), energy consumption (generation of $O_3$, for example), and unsatisfactory degradation efficiency (ultrasound alone, for example) [16].

Among AOPs methods, techniques that are based on UV irradiation have been widely studied. For example, photolytic/photocatalytic effects under different UV light sources, such as UVC and VUV, have been investigated [18–21]. Under VUV irradiation at 185 nm, water is homolyzed into a hydrogen atom and hydroxyl radicals. Other strong oxidants, such as ozone and hydrogen peroxide, are formed simultaneously [22]. However, commercially available VUV lamps have a main output at 254 nm with only a small amount (<10% of the total intensity of irradiation) at 185 nm. Therefore, semiconductors, such as titanium dioxide ($TiO_2$), can be used to produce more hydroxyl radicals via photocatalysis [23]. $TiO_2$ is a well-known photocatalyst, because of the advantages of a superior photocatalytic oxidation ability and optical and electronic properties [24]. Additionally, the $TiO_2$ photocatalyst is inexpensive, commercially available, non-toxic, and photochemically stable [25].

The efficacy of decontamination techniques is highly affected by the physical properties and exterior morphology of a surface, such as fresh produce [26]. Studies have also demonstrated that internalization and colonization of bacterial cells in damaged tissues or in niches on complex matrices also make removal from vegetables difficult [2]. In addition, several studies have reported that the reduction degree of pesticide residues varies, depending on the contact area, tissue structure, compactness, surface area, thickness, and the amount of wax on the cuticle [27,28]. Hence, a standardized model matrix with a regular size and shape and homogeneous surface characteristics is needed.

The objectives of this study included evaluation of the effects on the degradation of pesticides and microorganisms in aqueous solutions of UV photolysis and UV-TiO$_2$ photocatalysis under VUV and UVC light sources. The removal efficiencies of pesticides and microorganisms on model matrices prepared to represent fresh produce were also studied for their simultaneous decontamination.

## 2. Materials and Methods

### 2.1. Chemicals, Reagents, and Microbial Strains

Table 1 provides chemical names, structures, and physicochemical characteristics of the four tested pesticides. Commercial pesticide products for each pesticide component were purchased from Farm Hannong (Seoul, Korea; Maccani for pyraclostrobin), Sungbo Chemicals Co., Ltd. (Seoul, Korea; Collis for boscalid), and Syngenta Korea (Seoul, Korea; Saphire for fludioxonil and Amistar for azoxystrobin). The solvents used for HPLC analyses were purchased from Honeywell Burdick & Jackson (Muskegon, MI, USA). Hydrochloric acid was purchased from Daejung Chemicals & Metals Co., Ltd. (Shiheung, Korea). Analytical grade standards for pyraclostrobin, boscalid, fludioxonil, and azoxystrobin and all other chemicals were purchased from Sigma–Aldrich (St. Louis, MO, USA). The commercial pesticide products consisted of emulsifying agents and other pesticides (Maccani, dithianon 24% and pyraclostrobin 8%; Collis, boscalid 18.2%, and kresoxim-methyl 9.1%; Saphire, fludioxonil 20%; Amistar, azoxystrobin 10%).

*Escherichia coli* K12 (ATCC 43895) and *Saccharomyces cerevisiae* (ATCC 38661) were purchased from Korean Culture Center of Microorganisms (KCCM, Seoul, Korea). *E. coli* K12 was used as a surrogate bacterium of *E. coli* O157:H7. Difco™ culture media were purchased from Becton, Dickinson and Company (Franklin Lakes, NJ, USA).

**Table 1.** Chemical structure and physicochemical properties of pesticides.

| Pesticides | Pyraclostrobin | Boscalid | Fludioxonil | Azoxystrobin |
|---|---|---|---|---|
| Structure |  |  |  |  |
| IUPAC name | methyl N-{2-[1-(4-chlorophenyl)-1H-pyrazol3-yloxymethyl] phenyl}(N-methoxy) carbamate | 2-chloro-N-(4′-chlorobiphenyl-2-yl)nicotinamide | 4-(2,2-difluoro-1,3-benzodioxol-4-yl) pyrrole-3-carbonitrile | Methyl (2E)-2-(2-{[6-(2-cyanophenoxy)pyrimidin-4yl]oxy}phenyl)-3-methoxyacrylate |
| Molecular formula | $C_{19}H_{18}ClN_3O_4$ | $C_{18}H_{12}Cl_2N_2O$ | $C_{12}H_6F_2N_2O_2$ | $C_{22}H_{17}N_3O_5$ |
| Molecular weight | 387.82 | 343.21 | 248.18 | 403.38 |
| V.P | $2.6 \times 10^{-5}$ mPa (20 °C) | $7.2 \times 10^{-4}$ mPa (20 °C) | $3.9 \times 10^{-4}$ mPa (25 °C) | $1.1 \times 10^{-7}$ mPa (20 °C) |
| Solubility | In water 1.9 mg/L (20~25 °C). | In water 4.6 mg/L (20~25 °C) | In water 1.8 mg/L (20~25 °C) | In water 6 mg/L (20 °C) |

## 2.2. Photolytic and Photocatalytic Treatments

### 2.2.1. Preparation of Pesticide Solutions

Single pesticide solutions were prepared by diluting each commercial pesticide with Milli-Q water to achieve a final concentration of 20 ppm of each active compound. A cocktail solution was prepared by diluting the four pesticides with Milli-Q water in a bowl to achieve 5 ppm of each active compound, corresponding to total pesticide concentrations of 20 ppm (Figure 1). The pesticide concentration range that was applied in this study was set up taking into account that the MRLs of small fruits and vegetables are in the range of approximately 1–20 ppm [29].

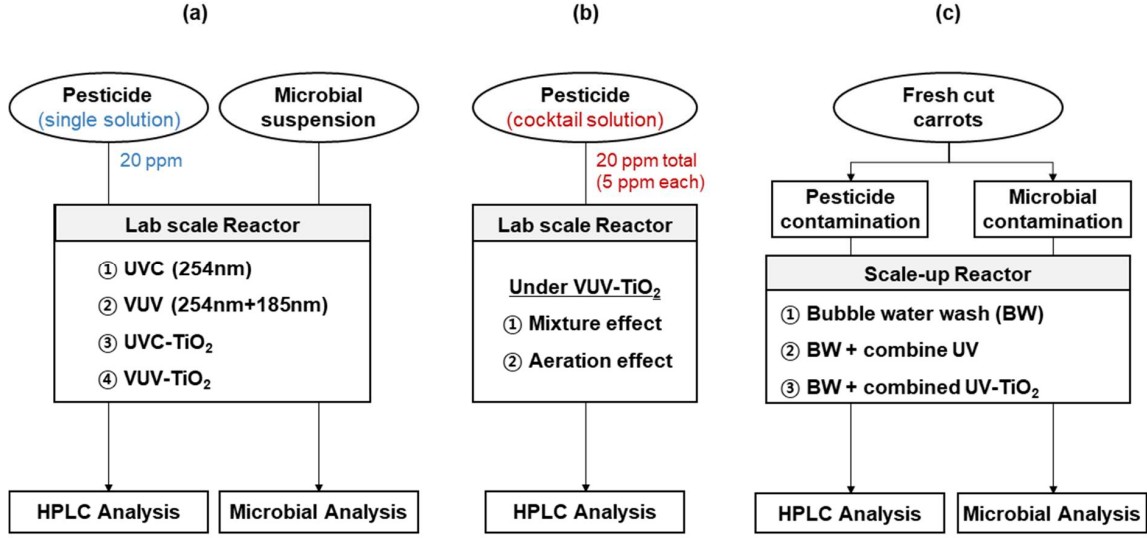

**Figure 1.** Experimental flow charts. (**a**) experiments to investigate the effect of the UV light source and TiO$_2$ photocatalysis on degradation of pesticides and microorganisms. (**b**) Experiments to investigate the effects of mixture compounds and aeration condition on the degradation efficiency of pesticides cocktail under VUV-TiO$_2$. (**c**) Experiments to investigate the removal effect of residual pesticide and microorganisms from carrot samples.

### 2.2.2. Preparation of Microbial Suspensions

Stock cultures of *E. coli* K12 were incubated in 100 mL of tryptic soy broth (TSB) at 37 °C for 24 h while shaking at 200 rpm. *S. cerevisiae* suspensions were prepared in 200 mL of potato dextrose broth after incubation at 25 °C for 48 h with shaking at 200 rpm. The cell suspensions of each of the microbial strains were washed by centrifugation at 4000 g for 10 min in a saline solution. The cell pellet of each strain was suspended in an equal volume (1500 mL) of the saline solution.

### 2.2.3. Degradation Treatments for Pesticides and Microorganisms in Aqueous Solutions

A lab-scale reactor was used to investigate the effects of the UV light source and the presence of TiO$_2$ photocatalyst on degradation of pesticides and microorganisms in aqueous solutions. The reactor consisted of a stainless-steel cylindrical vessel with an effective volume of 4.0 L. Two types of low-pressure mercury lamps were used to provide UVC and VUV irradiation. A UVC lamp capable of emitting light of a wavelength of 254 nm (35 W, 25 mW/cm$^2$, Sankyo Denki, Japan) and a VUV lamp emitting at both 254 nm and 185 nm (35 W, 25 mW/cm$^2$, Sankyo Denki, Japan) were used. The lamps were placed in the center of the reactor with quartz tube protection. A TiO$_2$-coated quartz tube (38 cm length, 24.5 mm outer diameter; TiO$_2$ coating thickness 0.7–0.9 mm; Taekyeong UV Co., Namyangju, Korea) was used for photocatalytic treatments, and an uncoated quartz tube was used for photolytic treatments. Four photolytic and photocatalytic treatments were applied: (i) UVC (254 nm) photolytic

treatment, (ii) VUV (254 nm and 185 nm) photolytic treatment, (iii) UVC-assisted $TiO_2$ photocatalytic treatment (UVC-$TiO_2$), and (iv) VUV-$TiO_2$ photocatalytic treatment (VUV-$TiO_2$).

An experiment was conducted in order to investigate mixture (or multi-component) effect on degradation efficiency. The degradation efficiency of each active compound in a cocktail solution (5 ppm of each active compound, corresponding 20 ppm of total active compounds) was compared to a single pesticide solution containing 20 ppm of each pesticide under VUV-$TiO_2$ (Figure 1).

Another experiment was conducted to investigate the effect of aeration under VUV-$TiO_2$ on the degradation of pesticides in a cocktail solution. This aeration was produced while using a magnetic stirrer only (no aeration), an air injector with magnetic stirring at 300 rpm (mild aeration) (SH-A2 bubble generator, Zhongshan Canhu Electric Factory, Zhongshan, China), and an air compressor with magnetic stirring (extra aeration) (2522C-10, Welch, Concord, MA, USA). Under mild aeration, air was supplied to the vessel through a silicone tube at a flow rate of 1.8 L/min. In extra aeration, compressed air was introduced through the micropore mesh plate installed at the bottom of the vessel at a flow rate of approximately 20 L/min. During photolytic and the photocatalytic treatments, 10 mL aliquots were taken for HPLC analysis at treatment times of 30, 60, 120, and 180 min.

Microbial inactivation experiments were conducted under UVC, VUV, UVC-$TiO_2$, and VUV-$TiO_2$ treatments (Figure 1). Briefly, each liquid culture of *E. coli* K12 and *Saccharomyces cerevisiae* was added to a sterile 0.85% NaCl solution in the lab-scale reactor. The total treated volume of microbial culture was 4 L. Liquid samples of approximately 10 mL were withdrawn from the reactor at 30 s intervals for up to 120 s.

### 2.3. Photolytic and Photocatalytic Treatments for Residual Pesticides and Microorganisms on Fresh Cut Carrots

### 2.3.1. Preparation of Fresh Cut Carrots as a Fresh Produce Surface Model

Fresh carrots were purchased from local markets in Seoul, Korea. Only the inner part of the carrot was used to exclude any existing pesticides and microflora. The inner part of the carrot was cut into a cylindrical shape using a cork borer with a diameter of 21.5 mm, a height of 20 mm, and a surface area of 20.76 $cm^2$. The cylindrical carrot models having a regular shape and size were then rinsed with running tap water for 1 min in order to reduce surface contamination and were stored at 4 °C until use within 24 h of all treatments.

### 2.3.2. Artificial Contamination of Fresh Cut Carrots with Pesticides and Microorganisms

Fresh cut carrot samples were immersed into previously prepared cocktail solutions for 30 min at ambient temperature using a shaking incubator at 80 rpm, followed by air drying for 4 h in a clean bench to ensure pesticide attachment to the model surface.

The carrot samples were treated with UV light (at a 30 cm distance from the light source) in a clean bench (model VS-1400LS, Vision Scientific, Bucheon, Korea) for 30 min to reduce amounts of naturally existing microflora. The samples were then immersed in each microbial inoculum solution and agitated by stirring in a shaking incubator at 80 rpm for 1 h. Inoculated samples were drained on sterile paper to remove excess inoculum and air-dried on a clean bench for 2 h to allow microbial attachment to the model surface.

### 2.3.3. Degradation Treatments for Residual Pesticides and Microorganisms Inoculated on Fresh Cut Carrots

A scale-up reactor was used to study the degradation efficiency of residual pesticides and microorganisms inoculated on fresh cut carrots. This reactor consisted of a stainless-steel vessel with a maximum volume of 50 L. Four UVC and four VUV lamps were alternately arranged at the top and bottom of the vessel. Low-pressure mercury lamps were placed in quartz tubes. A $TiO_2$-coated quartz tube was used for photocatalytic treatments and a $TiO_2$-uncoated quartz tube was used for photolytic treatments. An air pump (Ho Hsing Ring Compressor RB30-510, H. S. Machinery Co.,

Ltd., New Taipei City, Taiwan) was fitted at the bottom of the reactor to create turbulent flow in the wash water to achieve random rotation of carrot samples for uniform treatment of surfaces and to supply sufficient oxygen during photolytic and photocatalytic treatments. The following treatments were carried out: (i) only bubble water wash in dark condition as a control (bubble water wash); (ii) a combined photolytic treatment of both UVC and VUV (combined UV) with bubble water wash; and, (iii) a combined UVC-TiO$_2$ and VUV-TiO$_2$ photocatalytic treatment (combined UV-TiO$_2$) with bubble water wash. Samples were treated at room temperature in a vessel containing 45 L of tap water.

### 2.4. Analysis of Pesticides

Treated carrot samples were homogenized using a Waring blender in an ammonium acetate-acetic acid solution (20 mM) prepared in methanol-water (95:5) followed by sonication for 5 min and centrifugation at 3500 rpm for 10 min to extract pesticides from the samples. The supernatant of extracts and the aliquot from the aqueous solution in the lab-scale reactor were taken for HPLC analysis performed on an Agilent 1260 Infinity HPLC system (Agilent Technologies, Santa Clara, CA, USA) with a Diode Array Detector. After injecting 10 μL of sample, the separation was performed in a ODS Hypersil C18 250 × 4.6 mm, 5.0 μm, column (Thermo Fisher Scientific, Waltham, MA, USA). A mobile phase system of (A) water and (B) acetonitrile was used. A multi-step gradient of 20% A and 80% B for 0 min; 80% A and 20% B for 40 min; and, 20% A and 80% B for 50 min was used. The flow rate was 1.0 mL/min, the temperature was 30 °C, and the wavelength was set at 270 nm.

### 2.5. Kinetic Study of Pesticide Degradation

The reaction kinetics of each experiment was investigated while using the reaction rate laws. The concentration at a given time versus time plots ([C] vs. time), the logarithm of the ration of the initial concentration (C$_0$) to the concentration at a given time versus time plots (ln ([C$_0$]/[C]) vs. time), and the reciprocal of the concentration versus time plots (1/[C] vs. time) were prepared to evaluate the mostly fit rate law of each pesticides degradation reaction in single solution and cocktail solution under VUV-TiO$_2$.

The graph that is linear implies the order of the degradation reaction with respect to each pesticide (the corresponding R$^2$ of linear regression indicates the fitness of the particular kinetics model). Subsequently, the slope of which upon linear regression equals the rate constant of reaction; pseudo zero order kinetics (K$_0$, mg L$^{-1}$ time$^{-1}$), pseudo first order kinetics (K$_1$, time$^{-1}$), and pseudo second order kinetics (K$_2$, L mg$^{-1}$ time$^{-1}$).

### 2.6. Microbiological Analysis

The total plate count method was used to enumerate viable microbial cells. Liquid samples before and after treatment to aqueous solution were serially diluted using solution, followed by inoculation of 1.0 mL of the solution onto duplicate plates containing an appropriate agar. For carrot samples, fresh cut carrots before and after treatment were homogenized in a stomacher (MIX 2, AES Laboratories, Combourg, France) in 80 mL of a sterile 0.85% NaCl solution for 2 min, and then serially diluted using the saline solution, followed by inoculation of 1.0 mL of the solution onto duplicate plates containing an appropriate agar.

Tryptic soy agar was used to detect *E. coli* K12 after incubation at 37 °C for 24 h. Potato dextrose agar was used to detect *S. cerevisiae* after incubation at 25 °C for 48–72 h. Colonies were counted in each case at the end of the incubation period. The detection limit was 10 CFU per plate, which equated to 8 CFU/cm$^2$. The log (N/N$_0$) value was calculated to determine the inactivation effect, where N$_0$ was the initial microbial count in samples before treatment and N was the viable microbial count after inactivation treatments.

### 2.7. Statistical Analysis

All of the experiments were performed in triplicate, and the data were presented as the mean ± standard deviation. Statistical analysis was performed by one-way analysis of variance (ANOVA) and Duncan's multiple range test with significance defined at $P < 0.05$ using the Statistical Package for the Social Sciences (SPSS version 24, IBM Corporation, Armonk, NY, USA).

## 3. Results and Discussion

### 3.1. Effects of the UV Light Source and TiO$_2$ Photocatalysis on Degradation of Pesticides

VUV was more effective than UVC for pesticide degradation in single solutions (20 ppm) of each pesticide (pyraclostrobin, boscalid, fludioxonil, and azoxystrobin) (Figure 2). Degradation percentage of boscalid, fludioxonil, and azoxystrobin were 28%, 87%, and 53% under VUV, and 3%, 58%, and 48% under UVC, respectively, after 180 min of treatment. Pyraclostrobin was eliminated after almost 60 min under both VUV and UVC.

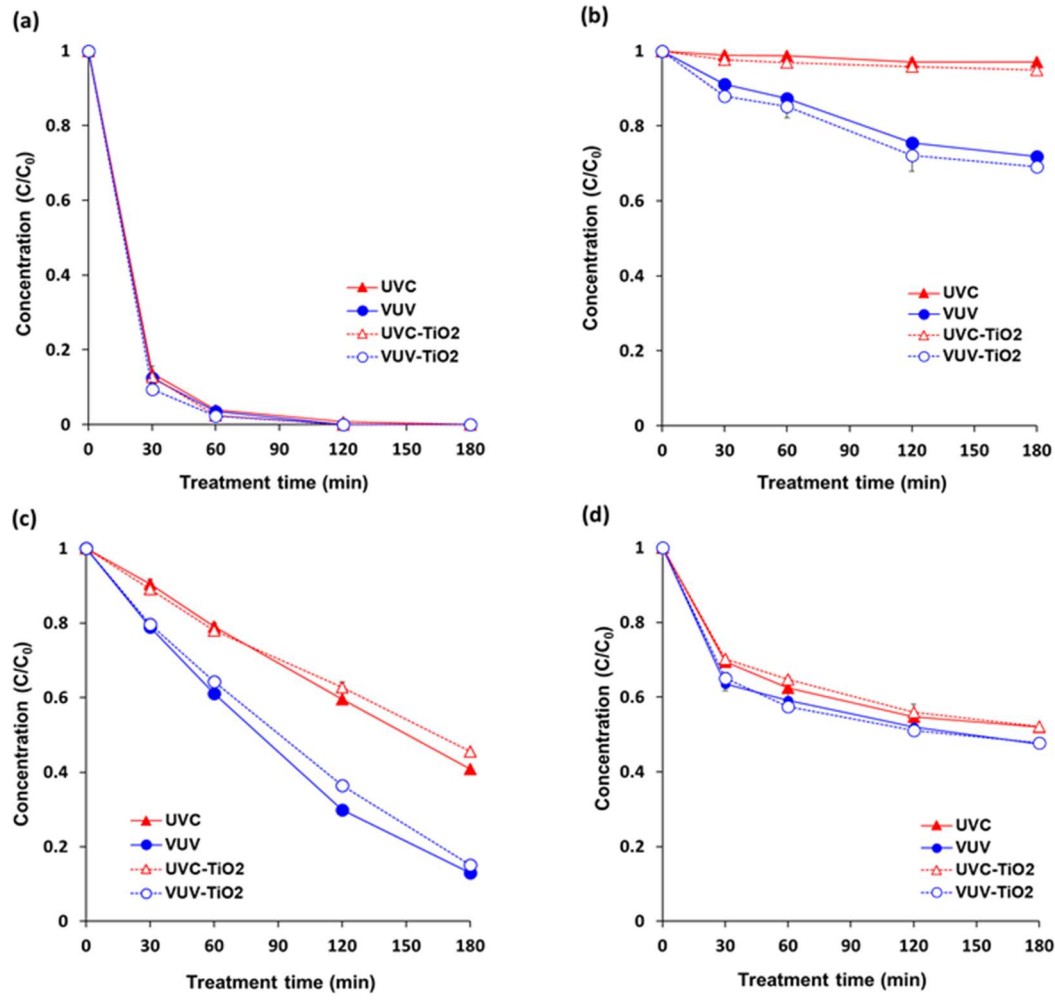

**Figure 2.** Degradation in individual pesticide solutions under different treatments. (**a**) pyraclostrobin; (**b**) boscalid; (**c**) fludioxonil; and, (**d**) azoxystrobin. The concentration of each single solution was 20 ppm (*w/v*). Sample solutions were continuously stirred with a magnetic stirrer at 300 rpm during treatment. UVC = UVC (254 nm) photolytic treatment; VUV = VUV (254 nm and 185 nm) photolytic treatment; UVC-TiO$_2$ = UVC-assisted TiO$_2$ photocatalytic treatment; VUV-TiO$_2$ = VUV-assisted TiO$_2$ photocatalytic treatment. Results represent the mean of three measurements ± standard deviation.

Pesticides generally have several functional parts possibly participating in photolysis. Based on a chemical class or a functional moiety basis, mechanisms in the direct photolysis of pesticides include photo-induced cleavage of a specific bond, intramolecular bond formation, rearrangement, and oxidation/reduction [14]. It was suggested that the photodegradation of pyraclostrobin was obtained by substituting chloride for an –OH group followed by the scission of the 4-hydroxyphenyl and pyrazol bond (N-C bond) [30]. For boscalid, it was proposed that the transformation pathways under UV irradiation was caused by ionizations at the site of double bonds in the pyridine ring as well as at the oxygen and the nitrogen atoms of the amide function [31]. Direct photolysis of fludioxonil was proposed to proceed through an initial radical cation intermediate resulting from photoionization from a singlet excited state [32]. Furthermore, geometrical isomerization such as the twisted structure around the C=C bond was theoretically proposed for photo-transformation pathways of azoxystrobin [33].

In particular, boscalid was hardly degraded by UVC (254 nm). Usually, boscalid is hydrolytically and photolytically stable in soil and water [34]. According to one report of Food Safety Commission (2004), 99.4–99.5% of the initial amounts of boscalid in buffer solution was detected after 30 days at 25 °C. In addition, 94.4% and 99.6% was detected under Xenon light (315–400 nm) after 15 days in buffer solution and under Xenon light (290–800 nm) after 120 h in sterilized water, respectively [35]. However, the result of the present study showed that VUV (185 nm) treatment was somewhat effective in degrading boscalid in an aqueous solution. A study also reported the photolytic degradation of boscalid under UV-visible light (200–650 nm) [31]. It can be considered that shorter wavelength can accelerate the rate of photolytic degradation. However, incidentally, maximum absorption wavelength of boscalid is 207 and 228 nm [36], which is shorter than UVC wavelength (254 nm) and longer than VUV wavelength (185 nm). It can be assumed that the degradation rate of photolysis is faster when the wavelength of the photon is shorter than the maximum absorption wavelength of the substance. Furthermore, VUV at 185 nm can produce energetic photons, leading to mainly direct photolysis of target compounds as well as indirect decomposition through the formation of hydroxyl radicals and other strong oxidants, such as ozone and hydrogen peroxide [23].

There was no significant difference between rate constants for UVC and UVC-$TiO_2$ treatments and between rate constants for VUV and VUV-$TiO_2$ treatments, attributed to a filtering effect of $TiO_2$ particles on the surface of the quartz tube that weakened the intensity of UV light. Therefore, no matter what oxidant was produced by $TiO_2$ photocatalysis, the direct UV photolysis effect was decreased due to a reduction in UV intensity that was caused by the $TiO_2$ coating. Several studies reported high formation rates of hydroxyl radicals (OH·) produced by UV-$TiO_2$ photocatalysis probably affected degradation of the pesticides and mineralization of pesticide intermediates [37,38]. However, further supporting investigation, such as TOC (total organic carbon) observation, will be needed in order to validate the $TiO_2$ photocatalytic degradation of the pesticides.

The degradation efficiency of an individual pesticide (5 ppm) in a cocktail solution (20 ppm) was not equal to the degradation efficiency in a single pesticide solution containing 20 ppm of each pesticide under VUV-$TiO_2$. After 180 min of each treatment, degradation percentages of pyraclostrobin, boscalid, and fludioxonil in the cocktail solution were significantly lower versus for single solutions (94.3% vs. 100%, 5.2% vs. 30.8%, 17.3% vs. 84.8%, respectively) (Figure 3). However, the degradation of azoxystrobin in a cocktail solution and in a single solution did not differ significantly during the whole treatment time. This was probably due to an inhibitory effect of each pesticide, its oxidative intermediates, and emulsifier additives in commercial pesticide brands. The $TiO_2$ photocatalytic process under VUV illumination relies primarily on a non-selective reaction of the strong oxidation potential of reactive oxygen species (ROS), such as hydroxyl radicals (OH·) and superoxide radical ion ($O_2 \cdot^-$), produced on the $TiO_2$ surface. The higher adsorption affinity of non-target compounds onto the $TiO_2$ surface can scavenge produced ROS and adsorb to the $TiO_2$ photocatalyst surface where they both displace the adsorption of target contaminants and interfere with the ROS production process [39]. In this study, azoxystrobin and its oxidative intermediates were probably degraded more by the VUV-$TiO_2$ photocatalytic treatment in competition, as compared with the other pesticides [40].

Table 2 shows the degradation rate constants of pyraclostrobin, boscalid, fludioxonil, and azoxystrobin. In single solutions, degradation rates of pyraclostrobin and boscalid were fitted to pseudo-first order kinetics, the rate of fludioxonil was fitted to pseudo zero order kinetics, and the rate of azoxystrobin was fitted to pseudo second order kinetics. Generally, pseudo-first order kinetics of experiments involving $TiO_2$ photocatalysis are an indicator of the Langmuir–Hinshelwood mechanism, a widely reported surface reaction mechanism [41]. The pseudo-second order kinetic model implies that the oxidation rate rapidly slowed during the course of the reaction as a result of accumulation of recalcitrant oxidation intermediates. The pseudo-zero order kinetic model of photolysis/photocatalysis indicated that there was no accumulation of intermediates resistant to further oxidation [19]. In the cocktail solution, the order of the degradation rates of pyraclostrobin, boscalid, and fludioxonil were changed to pseudo-second order kinetics, which indicated that the presence of azoxystrobin was the causative factor that decreased the degradation rates of pyraclostrobin, boscalid, and fludioxonil in the cocktail solution due to the accumulation of oxidative intermediates of azoxystrobin [42].

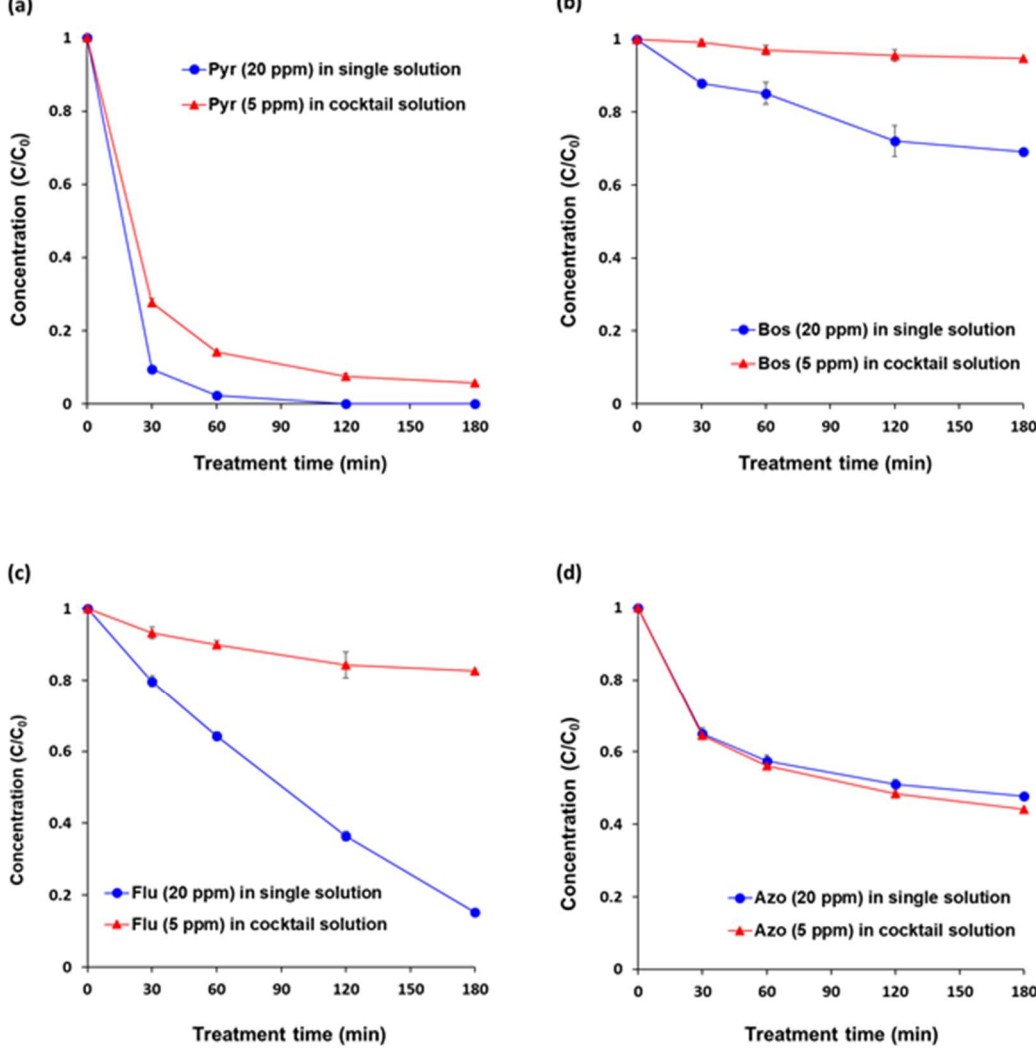

**Figure 3.** Degradation efficiencies of the VUV-assisted $TiO_2$ photocatalytic treatment (VUV-$TiO_2$) in a cocktail solution of pesticides. (**a**) pyraclostrobin (Pyr), (**b**) boscalid (Bos), (**c**) fludioxonil (Flu), and (**d**) azoxystrobin (Azo). The pesticide concentration of each single solution was 20 ppm (*w/v*). The concentration of each pesticide in a cocktail solution was 5 ppm (*w/v*) and the total pesticide concentration of the cocktail solution was 20 ppm (*w/v*). All of the treatments were performed under magnetic stirring at 300 rpm. Results represent the mean of three measurements ± standard deviation.

**Table 2.** Degradation rate constants in single solutions and in a cocktail solution. $K_0$ (mg L$^{-1}$ min$^{-1}$), $K_1$ (min$^{-1}$), $K_2$ (L mg$^{-1}$ min$^{-1}$), and the corresponding $R^2$.

| Rate Constant | Pyraclostrobin | | | Boscalid | | | Fludioxonil | | | Azoxystrobin | | |
|---|---|---|---|---|---|---|---|---|---|---|---|---|
| | $K_0$ ($R^2$) | $K_1$ ($R^2$) | $K_2$ ($R^2$) | $K_0$ ($R^2$) | $K_1$ ($R^2$) | $K_2$ ($R^2$) | $K_0$ ($R^2$) | $K_1$ ($R^2$) | $K_2$ ($R^2$) | $K_0$ ($R^2$) | $K_1$ ($R^2$) | $K_2$ ($R^2$) |
| Single solution | 0.4888 (0.8051) | 1.8978 (0.9809) | 21.753 (0.9051) | 0.0775 (0.9560) | 0.0935 (0.9600) | 0.1141 (0.9590) | 0.213 (0.9926) | 0.4556 (0.9092) | 1.2681 (0.7503) | 0.1184 (0.7920) | 0.1719 (0.8635) | 0.2606 (0.9263) |
| Cocktail solution | 0.2088 (0.6975) | 0.7035 (0.9394) | 4.2827 (0.9787) | 0.0014 (0.9753) | 0.0144 (0.9759) | 0.0148 (0.9765) | 0.0435 (0.9650) | 0.0480 (0.9714) | 0.0531 (0.9763) | 0.1278 (0.8221) | 0.1921 (0.8992) | 0.3043 (0.9591) |

Herein, the VUV-TiO$_2$ treatment seemed to be closely related to selective reaction oxidation mechanisms. Ozone is generated by VUV light (mainly at 185 nm) that is emitted through the uncoated portion of the TiO$_2$ coated quartz tube. Ozone, which is based on a resonance hybrid structure, can act as a 1,3-dipole, an electrophilic agent, and a nucleophilic agent during reactions [43]. These three types of reactions occur quickly in solutions of organic matter that contain double bonds, activated aromatic groups, and amines [44]. Thus, depending on the molecular structural characteristics of azoxystrobin, pyraclostrobin, boscalid, and fludioxonil (Table 1), they are differently favored for ozone generated during VUV-TiO$_2$ photocatalysis treatment, which results in a competitive reaction [45,46].

### 3.2. Effects of Aeration on the Pesticide Degradation Efficiency

Aeration enhanced degradation rates of the pesticides in cocktail solution under VUV-TiO$_2$ (Figure 4). Mild aeration for 3 h degraded pesticides by 96.4%, 14.7%, 29.6%, and 63.5% for pyraclostrobin, boscalid, fludioxonil, and azoxystrobin, respectively, indicating a better effect when compared to no aeration which degraded pesticide by 94.3%, 5.2%, 17.3%, and 55.8%. Extra aeration further enhanced the degradation efficiency with 100.0%, 21.1%, 56.9%, and 80.9% degradation for pyraclostrobin, boscalid, fludioxonil, and azoxystrobin, respectively. These increased degradation percentages under extra aeration can be attributed to the reactor with an aeration system that supplied compressed air at a high flow rate through the stainless micropore mesh at the bottom of the vessel. Additionally, a fresh supply of oxygen in water through the pump might have enhanced the oxidation of organic compounds [47,48]. In addition, it is estimated that air bubbles in the pesticide solution also reduced the heterogeneity of photolysis by 185 nm light. In the VUV-TiO$_2$ treatment, the 185 nm light of the VUV lamp was emitted through the uncoated portion of the TiO$_2$ coated quartz tube, and the path that the 185 nm light that penetrated the water was very short due to a high absorption coefficient. Therefore, the intensity of 185 nm light was decreased nearly 90% in a 5.5 mm thickness of water. For this reason, under VUV-TiO$_2$ treatment, hydroxyl radicals (OH·) were only generated in a thin layer near the quartz tube [49,50]. It was estimated that 185 nm light penetrated air bubbles in contact with the TiO$_2$ coated tube, generating hydroxyl radicals (OH·) in the expanded layer, which contributes to an increase in the pesticide degradation efficiency by overcoming heterogeneous photolysis.

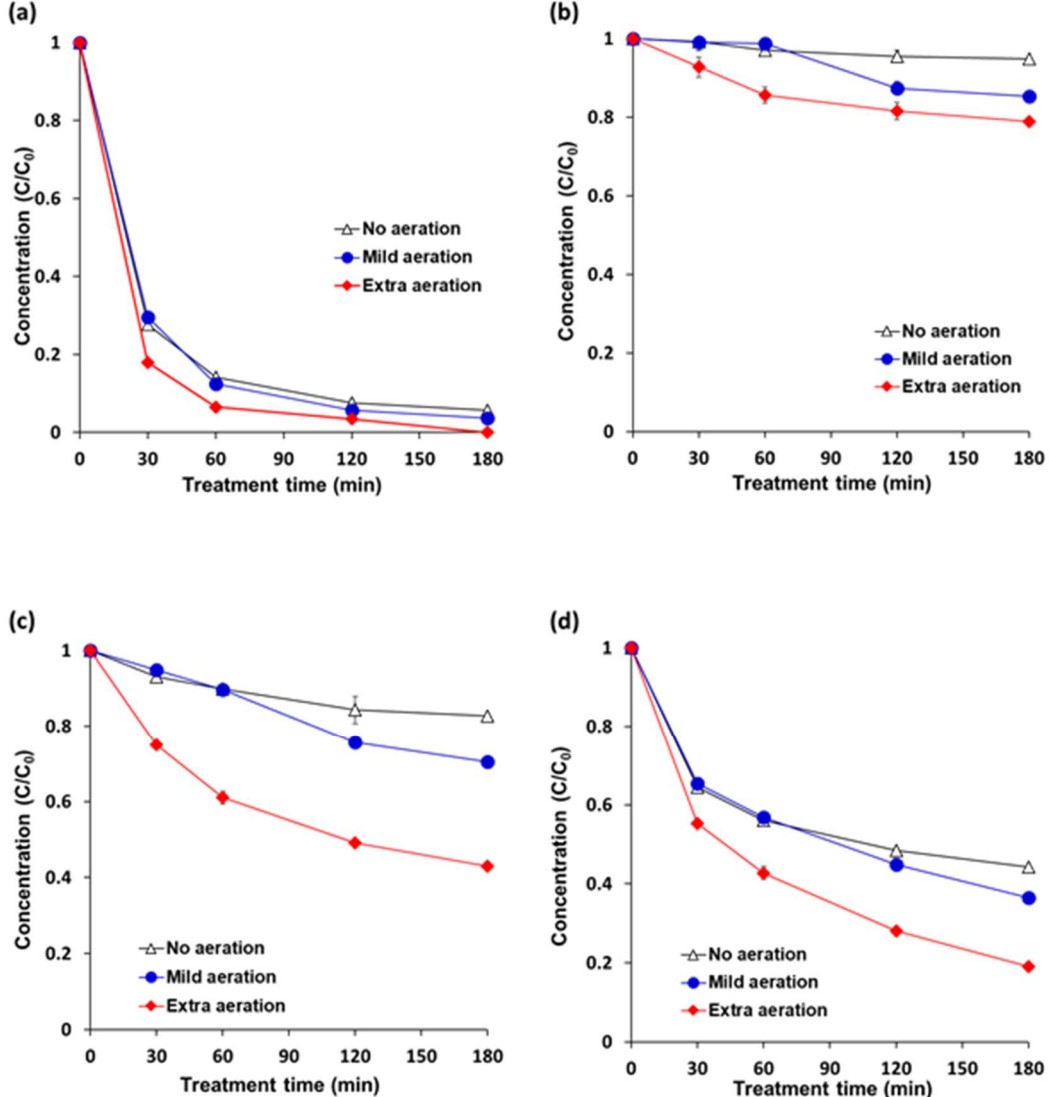

**Figure 4.** Effects of air bubble induced agitation on the pesticide degradation efficiency under VUV-assisted $TiO_2$ photocatalytic treatment (VUV-$TiO_2$). (**a**) pyraclostrobin; (**b**) boscalid; (**c**) fludioxonil; and (**d**) azoxystrobin. A cocktail solution was composed of individual pesticides at 5 ppm (*w/v*). No aeration = magnetic stirring at 300 rpm; mild aeration = air flow rate of 1.8 L/min with magnetic stirring at 300 rpm; and, extra aeration = air flow rate of 20 L/min with magnetic stirring at 300 rpm. Results represent the mean of three measurements ± standard deviation.

### 3.3. Effects of the UV Light Source and TiO₂ Photocatalysis on Inactivation of Microorganisms

The initial population of *E. coli* K12 and *S. cerevisiae* were 6.9 log CFU/mL and 5.3 log CFU/mL, respectively. For *E. coli* K12, VUV treatment showed a slightly higher inactivation effect than UVC (Figure 5a). For the first 30 s, UVC-$TiO_2$ and VUV-$TiO_2$ resulted in bacterial reductions of 0.9 log and 1.5 log CFU/mL, respectively, which were slightly lower than for UVC and VUV alone. After 120 s of treatment, UVC-$TiO_2$ and VUV-$TiO_2$ resulted in bacterial inactivation of 4.0 log and 3.9 log CFU/mL, respectively, which were slightly higher than for UVC and VUV alone.

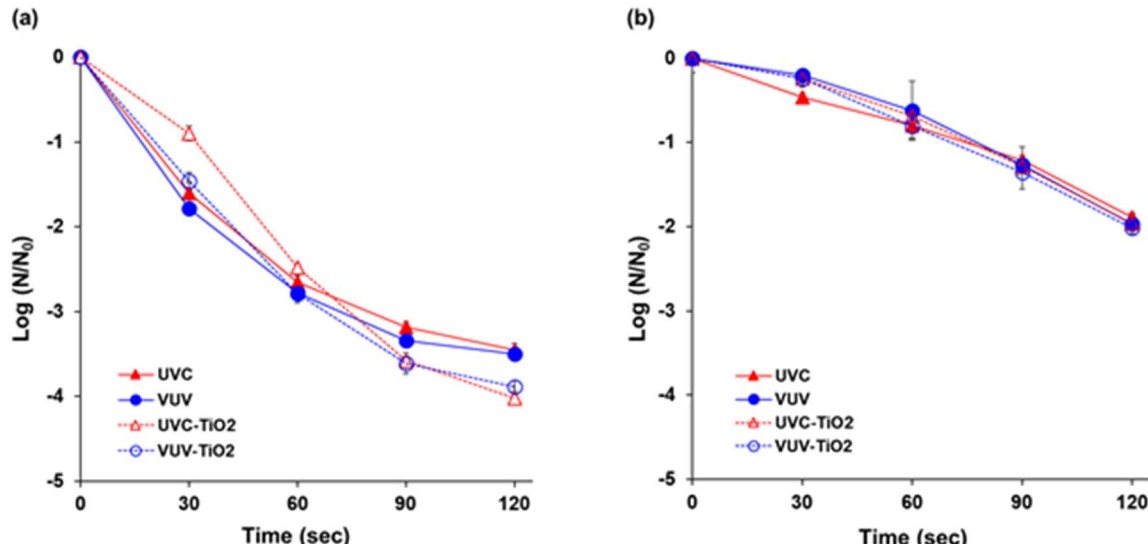

**Figure 5.** Effects of the UV light source and TiO$_2$ photocatalytic on inactivation of microorganisms in aqueous media. (**a**) *E. coli* K12; (**b**) *Saccharomyces cerevisiae*. UVC = UVC (254 nm) photolytic treatment; VUV = VUV (254 nm and 185 nm) photolytic treatment; UVC-TiO$_2$ = UVC-assisted TiO$_2$ photocatalytic treatment; VUV-TiO$_2$ = VUV-assisted TiO$_2$ photocatalytic treatment. The $N_0$ of *E. coli* K12 and *S. cerevisiae* were 6.9 and 5.3 log CFU/mL, respectively. All of the treatments were performed under extra aeration. Results represent the mean of three measurements ± standard deviation.

For *S. cerevisiae*, UVC, VUV, UVC-TiO$_2$, and VUV-TiO$_2$ showed similar inactivation effects of 0.2–0.4 log, 0.6–0.8 log, 1.2–1.3 log, and 1.9–2.0 log CFU/mL for 30, 60, 90, and 120 s, respectively (Figure 5b).

In this study, UVC, VUV, UVC-TiO$_2$, and VUV-TiO$_2$ provided similar inactivation effects against microorganisms in aqueous medium. It was estimated that UV light at 254 nm, which was used in all treatment conditions (UVC, VUV, UVC-TiO$_2$, and VUV-TiO$_2$), was a key factor for microbial inactivation. UVC spectrum, especially at 250–270 nm, is the most lethal wavelength range for microorganisms due to strong absorption by nucleic acids. The UV-induced damage to the DNA and RNA of a microorganism is due to the dimerization of pyrimidine molecules. In particular, when thymine (found only in DNA) is dimerized, it becomes difficult to replicate nucleic acids and, even if replication occurs, it often produces a defect that interferes with the survival of microorganisms [51]. In addition, the advantage of the TiO$_2$ photocatalytic reaction that overcomes the shadow effect, the limitation of UV irradiation, may not be clearly revealed in a short time for microorganisms in a transparent aqueous solution. Further investigation is needed to uncover any reactive difference between photolysis and TiO$_2$ photocatalysis. It can be achieved using UV irradiation, such as UVB or UVA, not to give serious damage to nucleic acids. However, because the objective of this study was to investigate the potential of photolysis and TiO$_2$ photocatalysis for simultaneous degradation of pesticides and microorganisms, VUV, and UVC spectrum with energetic photon, which is effective for the degradation of chemical compounds were chosen as a light source.

In this study, Gram-negative *E. coli* K12 showed less resistance to the TiO$_2$ photocatalytic treatments (UVC-TiO$_2$ and VUV-TiO$_2$) than yeast *S. cerevisiae*. Previous studies attributed this behavior to distinctive interactions of *E. coli* with TiO$_2$ surfaces [52–55]. Another study proposed that cell wall structures in microorganisms are responsible for different sensitivities towards TiO$_2$ photocatalysis and UV treatment [56]. Another study suggested that the susceptibility of microorganisms to reactive oxygen species can be generally arranged in the following descending order: viruses > prions > Gram-negative bacteria > Gram-positive bacteria > yeasts > molds [57].

### 3.4. Removal of Residual Pesticides on Fresh Cut Carrots under Combined UV and Combined UV-TiO₂ Treatments

Initial concentrations of pyraclostrobin, boscalid, fludioxonil, and azoxystrobin on carrot samples were approximately 3.5 ppm, 2.9 ppm, 3.4 ppm, and 5.7 ppm, respectively. An air pump that was fitted at the bottom of the reactor helped to create turbulence in the wash water for random rotation of carrots to achieve uniform degradation (Figure 6). This bubble generation system in the wash water also helped with the removal of residual pesticides. After 60 min of bubble water washing, the removal percentages of pyraclostrobin, boscalid, fludioxonil, and azoxystrobin were 53.0%, 74.2%, 68.5%, and 75.7% of the initial concentrations, respectively. Both combined UV and combined UV-TiO₂ showed removal percentages of approximately 75%, 72%, 83%, and 94%, respectively. Combined UV-TiO₂ did not produce a better photocatalytic removal efficiency than combined UV, similar to the data in Figure 2. No significant variations were observed in a trend of reduction for all measurements. Thus, fresh cut carrots can be a suitable simulation model for pesticide reduction experiments on fresh produce.

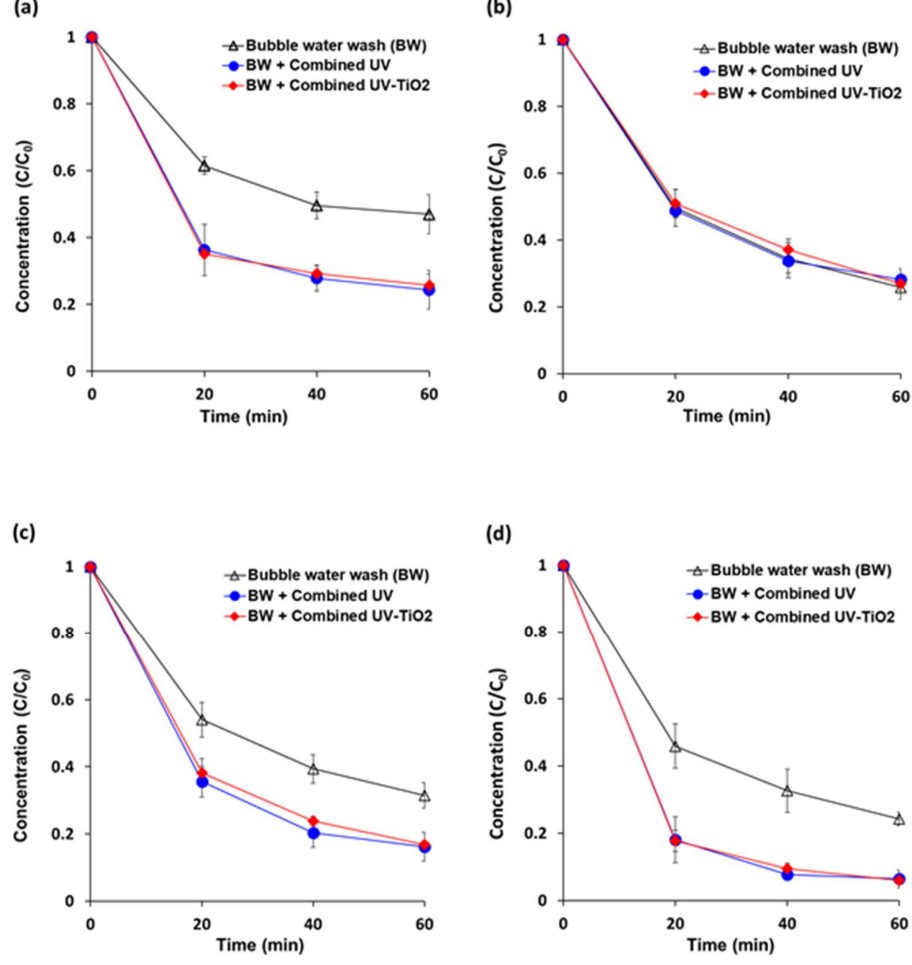

**Figure 6.** Removal of residual pesticide contamination from fresh cut carrots using combined UV and combined UV-TiO₂. (**a**) pyraclostrobin; (**b**) boscalid; (**c**) fludioxonil; and, (**d**) azoxystrobin. Bubble water wash (BW) = air bubbling as a control; Combined UV = UVC and VUV; combined UV-TiO₂ = UVC-assisted TiO₂ photocatalytic; and, VUV-assisted TiO₂ photocatalytic treatment. The $C_0$ of pyraclostrobin, boscalid, fludioxonil, and azoxystrobin were approximately 3.5, 2.9, 3.4, and 5.7 ppm, respectively. The results represent the mean of three measurements ± standard deviation.

In this study, the degradation rate of pesticides was slowed down along the course of the reaction time that can be attributed to the accumulation of oxidation intermediates. This decontamination technique for solid food materials is expected to be improved for a higher degradation rate by applying an open system with a continuous water flow through which clean water can be continuously supplied and treated wash water containing oxidizing intermediates is discharged. Meanwhile, it is necessary to first understand the presence and mutual effects of components that might reduce the reaction efficiency of the entire system, such as azoxystrobin in the present investigation, in order to apply it to the liquid medium of a multi-component system such as surface/ground water. Based on this information, this decontamination technique could be applied to multi-component systems in combination with mechanical means, including high turbulence, positive irradiation structures, and static mixers or bubbles.

### 3.5. Inactivation of Microorganisms on Fresh Cut Carrots with Combined UV and Combined UV-TiO$_2$ Treatments

Initial populations of *E. coli* K12 and *S. cerevisiae* on the carrot surface model were in the range of 6.5–8.0 log CFU/cm$^2$ and 4.3–5.6 log CFU/cm$^2$, respectively. The populations of *E. coli* K12 were reduced by 0.7 log, 1.3 log, and 1.5 log after 60 min of bubble water wash, combined UV, and combined UV-TiO$_2$ treatments, respectively (Figure 7a). For *S. cerevisiae*, 0.6 log, 1.2 log, and 1.6 log inactivations were achieved by the bubble wash, combined UV, and combined UV-TiO$_2$, respectively, for 60 min (Figure 7b). Combined UV-TiO$_2$ resulted in slightly more degradation than combined UV for all treatment times, but without a significant difference.

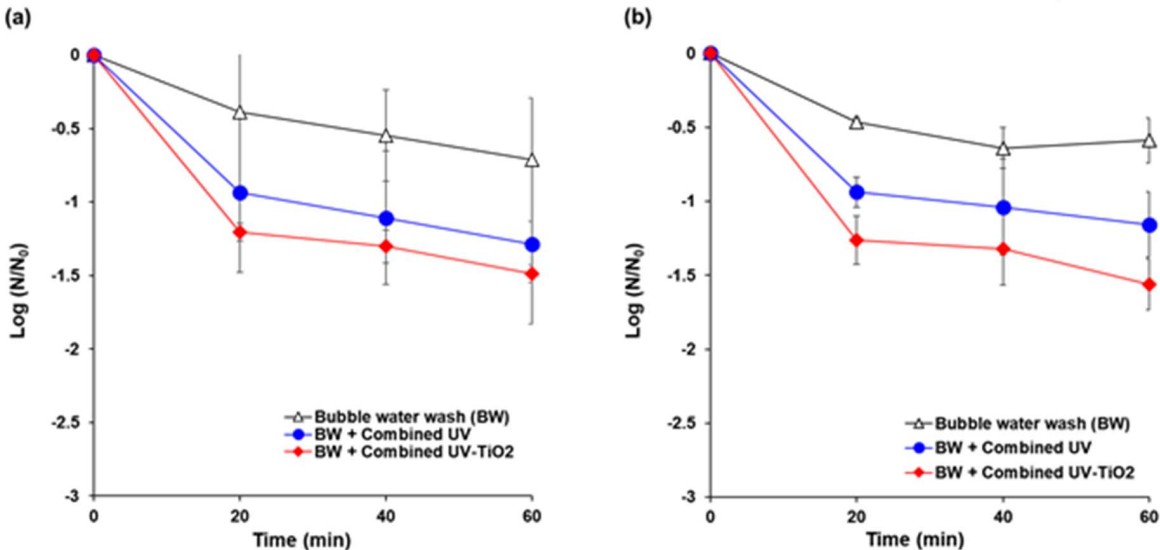

**Figure 7.** Inactivation of microbial contamination on fresh cut carrots using combined UV and combined UV-TiO$_2$. (**a**) *E. coli* K12; (**b**) *Saccharomyces cerevisiae*. Bubble water wash (BW) = air bubbling as a control; Combined UV = UVC and VUV; combined UV-TiO$_2$ = UVC-assisted TiO$_2$ photocatalytic; and, VUV-assisted TiO$_2$ photocatalytic treatment. The $N_0$ of *E. coli* K12 and *S. cerevisiae* were in the range of 6.5–8.0 log CFU/cm$^2$ and 4.3–5.6 log CFU/cm$^2$, respectively. The results represent the mean of three measurements ± standard deviation.

The slightly higher inactivation effect of combined UV-TiO$_2$ can be attributed to the decrease in shadowing effect in UV disinfection process. UV treatment is a conventional method for disinfection of fresh produce, but it has limitations for industrial applications due to the shadowing effect. Microorganisms on rough surfaces or complex matrices of food are difficult to target while using UV light, because target microorganisms on food surfaces must directly face a UV lamp for inactivation [58,59]. Furthermore, when UV light is commercially applied to fresh produce in a processing line, individual

object also casts shadow on other objects. In the past decades, research on UV disinfection has focused on hurdle technologies or reactor design, such as water-assisted UV system, to achieve higher inactivation efficacy [60–62]. In UV-TiO$_2$ photocatalytic treatment, TiO$_2$ absorbs the energy of photon under irradiation of UV light. When the energy of photon is higher than the width of the semiconductor's void band (3.2 eV; the UV light with $\lambda$ < 387.5 nm), the electron of valence band will transfer to the conduction band. Therefore, the pairs of electron-holes are created in the TiO$_2$, and the charge will be transferred between electron-hole pairs and adsorbed species (H$_2$O, OH-, O$_2$, other reactants) on the TiO$_2$, and then photo-oxidation occur. Finally, reactive oxygen species (OH·, HO$_2$, O$_2$·$^-$, H$_2$O$_2$), mainly hydroxyl radicals (OH·), are generated [63]. Hydroxyl radical can cause cell wall disruption and oxidation of cytoplasmic contents, eventually leading to microbial cell death [2,64,65]. Hydroxyl radical can reduce the shadow effects of the UV disinfection process, providing a far-reaching antimicrobial impact on bacterial cells [26]. UV-TiO$_2$ photocatalysis showed higher reduction in the natural microflora populations and inoculated populations of the pathogenic bacteria *E. coli*, *Listeria monocytogenes*, *Staphylococcus aureus*, and *Salmonella* Typhimurium on iceberg lettuce compared to UV alone and a sodium hypochlorite solution treatment [66].

In this study, tap water was used as wash water to exclude the effects of salts and minerals on microbial inactivation in TiO$_2$ photocatalytic treatment. For example, in TiO$_2$ photocatalysis, chloride ions (Cl$^-$) react with photogenerated holes from valence band of TiO$_2$ to form chlorine radicals (Cl·). Chlorine radicals then react with Cl$^-$ and convert into chloride radical anion (Cl$_2^-$·) [67]. For this reason, photocatalytic disinfection of bacteria in saline water may be more effective than the plain water. It was also considered that distilled or demineralized water and saline solutions are not suitable for applications to industry due to limitations, such as cost and additional equipment.

## 4. Conclusions

Fresh cut carrots proved their potential as a suitable model surface for evaluation of degradation of pesticides and microorganisms on the surface of fresh produce. VUV was more effective than UVC for pesticide degradation in aqueous solution; however, there was a little TiO$_2$ photocatalytic effect. Combined UV and combined UV-TiO$_2$ showed the efficient removal of residual pesticides from fresh cut carrot surfaces, but there was no significant difference between the treatments. UVC, VUV, UVC-TiO$_2$, and VUV-TiO$_2$ provided similar inactivation effects against microorganisms in aqueous medium. However, the combined UV-TiO$_2$ treatment showed slightly higher inactivation effects than combined UV on the surface of fresh cut carrots, which were attributed to a decrease in the shadowing effect under UV irradiation alone. In conclusion, photolytic and TiO$_2$ photocatalytic treatments under UV irradiation, including VUV as a light source, showed potential for the simultaneous degradation of pesticides and microorganisms. This study contributes to research focused on a non-chemical and residue-free decontamination technique for ensuring both microbiological and chemical safety of fresh produce.

**Author Contributions:** J.P. and D.-U.L. conceived the idea and designed experiments. S.W.C., J.U.K., D.-H.K., S.Y., and S.H.J. performed all experiments and interpreted data. S.W.C., H.M.S., and J.U.K. wrote the manuscript. J.P. and D.-U.L. revised and improved the manuscript. All authors have read and agreed to the published version of the manuscript.

**Funding:** This study was funded by Nutrex Technology Co., Seongnam, Republic of Korea.

**Conflicts of Interest:** The authors declare no conflict of interest.

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
