# Peer review of "Photolysis and TiO2 Photocatalytic Treatment under UVC/VUV Irradiation for Simultaneous Degradation of Pesticides and Microorganisms"

_applsci, doi:10.3390/app10134493_

Round 1
Reviewer 1 Report
I felt that the quality of this paper was drastically improved. In addition, I’m satisfied that the author(s) could understand the difference between photolysis and AOPs. Especially, I expect that the author(s) could understand that the efficiency of photolysis is very strong compare with AOPs for elimination of pesticides. Of course, using by VUV, water cleavage is also occurred and activated species derived from H2O is also act as decomposition of organic compounds (as AOPs). However, the author(s) also mentioned about this matter in this paper.
More improve this paper, I will suggest next two points:
- In the section of introduction, description about photolysis is nothing.In this paper, efficiency of photolysis against elimination of pesticides and bacteria is very much, therefore, description about photolysis must need at the same level of AOPs description (of course, references are also).
- Choice of tap water using is fair for face up to demonstration, however, the author(s) should add that reason why they used tap water instead of distilled water and/or saline water.
If the author(s) will respond to above suggestion, I will agree to publish this paper.
Author Response
I felt that the quality of this paper was drastically improved. In addition, I’m satisfied that the author(s) could understand the difference between photolysis and AOPs. Especially, I expect that the author(s) could understand that the efficiency of photolysis is very strong compare with AOPs for the elimination of pesticides. Of course, using VUV, water cleavage has also occurred and activated species derived from H2O is also act as decomposition of organic compounds (as AOPs). However, the author(s) also mentioned this matter in this paper.
è Thank you for your time and valuable inputs.
More improve this paper, I will suggest the next two points:
- In the section of the introduction, a description of photolysis is nothing. In this paper, the efficiency of photolysis against the elimination of pesticides and bacteria is very much, therefore, a description of photolysis must need at the same level of AOPs description (of course, references are also).
è Photolysis is one of the most important degradation mechanisms for pollutants such as pesticides. Photolytic methods are based on providing high energy to chemical compounds in the form of photon [J. Pestic. Sci. 2018, 43, 57-72]. Most pesticides show UV–Vis absorption bands at relatively short UV wavelengths. UV irradiation leads to the promotion of pesticides to their excited singlet states, and such excited states can then undergo homolysis, heterolysis, and/ or photoionization [J. Photochem. Photobiol. B, Biol. 2002, 67, 71-108]. However, sunlight reaching the Earth’s surface (mainly UVA, with varying amounts of UVB) contains only a very small amount of short wavelength UV radiation, the direct photolysis of pesticides by sunlight is generally expected to be limited. Thus, direct UV irradiation at short wavelength causes characteristic reactions such as bond scission, cyclization, and rearrangement, which are scarcely observed in hydrolysis and microbial degradation [J. Pestic. Sci. 2018, 43, 57-72]. The intensity and wavelength of the UV radiation or the quantum yield of the compound to be eliminated are factors that affect the performance of the photodegradation process. As a source of UV radiation, mercury vapor lamps are usually used [Water 2020, 12, 102]. This information is now included in the introduction section (lines 49-61).
- The choice of tap water using is fair for face up to the demonstration, however, the author(s) should add that reason why they used tap water instead of distilled water and/or saline water.
è Tap water was used to wash water in this study to exclude the effects of salts and minerals on microbial inactivation in TiO2 photocatalytic treatment. For example, in TiO2 photocatalysis, chloride ions (Cl-) react with photogenerated holes from the valence band of TiO2 to form chlorine radicals (Cl⋅). Chlorine radicals then react with Cl- and convert into chloride radical anion (Cl2-⋅) [Water Air Soil Pollut. 2014, 225, 1922]. For this reason, photocatalytic sterilization of bacteria in saline water may be effective more than plain water. It was also considered that distilled (or demineralized water) and saline solutions are not suitable for applications to the industry due to limitations such as cost and additional equipment. This information is now included in the introduction section (lines 186-187, 425-431).

Reviewer 2 Report
The paper entitled “Photolysis and TiO2 photocatalytic treatment under UVC/VUV irradiation for simultaneous degradation of pesticides and microorganisms” is an interesting article, well presented but for my opinion before that article is accepted minor revision will be addressed.
Abstract
Lines 14-15. (UVC-TiO2), for clarity the authors should report(UVC-titanium dioxide). Please check and revise it.
Lines 90-95. The authors should draw a flow chart in order to explains better and with more clearly the experimental design.
In the line 92 the authors write “…of each active compound” and in line 93 “… each active pesticide”. Do the active compound and active pesticide have the same meaning?
Lines 126-129. Indicate the total volume of microbial culture inside the vessel and their cells concentration. The authors should indicate the method used to count the cells before and after treatment used.
Line 184. Please indicate the model used in statistical analysis.
Lines 236-238. If the authors have conducted only this experiment with the pesticide’s cocktails, these should be reported in the material and methods section.
Lines 236-263. Why the authors did not mention the effect of treatment on the different concentrations of pesticides used (in particular 2.5 and 1.25 ppm)?
Line 281. The pesticides cocktails …please check and revise it.
Table 2. the authors should indicate in the M&M section how the degradation rate constants were calculated
All Figures. The figure should be self-explanatory for the reader. Please indicate the meaning of each abbreviations reported in the figures (for example figure 2. Pyraclostrobin (Pyr) and so on…
Author Response
Reviewer 2
Comments and Suggestions for Authors
The paper entitled “Photolysis and TiO2 photocatalytic treatment under UVC/VUV irradiation for simultaneous degradation of pesticides and microorganisms” is an interesting article, well presented but for my opinion before that article is accepted minor revision will be addressed.
- Thank you very much.
Abstract
C#1. Lines 14-15. (UVC-TiO2), for clarity the authors should report (UVC-titanium dioxide). Please check and revise it.
- We have clarified the statement as follows: “UVC-assisted titanium dioxide photocatalysis (UVC-TiO2), and VUV-assisted titanium dioxide photocatalysis (VUV-TiO2)” (line 14-15).
C#2. Lines 90-95. The authors should draw a flow chart in order to explains better and with more clearly the experimental design.
- We have added a flow chart as Figure 1 for better understanding of readers (line 123-129).
C#3. In the line 92 the authors write “…of each active compound” and in line 93 “… each active pesticide”. Do the active compound and active pesticide have the same meaning?
- Yes, they have same meaning. Single pesticide solutions were prepared by diluting each commercial pesticide with Milli-Q water to achieve a final concentration of 20 ppm of each active compound. A cocktail solution was prepared by diluting the four pesticides with Milli-Q water in a bowl to achieve 5 ppm of each active compound, corresponding to total pesticide concentration of 20 ppm. We have clarified the statement in the revised manuscript (line 116-117, Figure 1, 151-153, 292-294)
C#4. Lines 126-129. Indicate the total volume of microbial culture inside the vessel and their cells concentration. The authors should indicate the method used to count the cells before and after treatment used.
- The total treated volume of microbial culture was 4 L, and the initial population of E. coli K12 and S. cerevisiae were approximately 6.9 and 5.3 log CFU/mL, respectively (line 165-166, 365).
The total plate count method was used to enumerate viable microbial cells. Liquid samples before and after treatment to aqueous solution were serially diluted using solution, followed by inoculation of 1.0 ml of the solution onto duplicate plates containing an appropriate agar. Tryptic soy agar was used to detect E. coli K12 after incubation at 37 °C for 24 h. Potato dextrose agar was used to detect S. cerevisiae after incubation at 25 °C for 48-72 h. Colonies were counted in each case at the end of the incubation period (line 225-234).
C#5. Line 184. Please indicate the model used in statistical analysis.
- Statistical analysis was performed by one-way analysis of variance (ANOVA) and Duncan’s multiple range test (line 240-241).
C#6. Lines 236-238. If the authors have conducted only this experiment with the pesticide’s cocktails, these should be reported in the material and methods section.
- We have moved the statement to the Materials and methods section (line 150-153) and rephrase the statement in Results and discussion section (line 292-294).
C#7. Lines 236-263. Why the authors did not mention the effect of treatment on the different concentrations of pesticides used (in particular 2.5 and 1.25 ppm)?
- Thank you for the comment. Actually, we conducted another experiment to investigate the effects of the initial concentration of pesticides on the degradation efficiency. However, we didn’t cover it in this paper. We conducted the experiment only with a cocktail solution (20 ppm total) containing 5 ppm of each pesticide. We have corrected the Section 2.2.1 (line 116-117).
C#8. Line 281. The pesticides cocktails …please check and revise it.
- We have revised the statement as follows: “the pesticides in cocktail solution” (line 339)
C#9. Table 2. the authors should indicate in the M&M section how the degradation rate constants were calculated
- We have added the method for kinetic study of pesticide degradation as Section 2.5 (line 212-223)
C#10. All Figures. The figure should be self-explanatory for the reader. Please indicate the meaning of each abbreviations reported in the figures (for example figure 2.Pyraclostrobin (Pyr) and so on…
- We have revised all figure legends by providing the meaning of each abbreviations and detail information (line 279-281, 324-328, 358-362, 377-380, 431-433, 488-491).

Reviewer 3 Report
Manuscript applsci-851060 by the authors of Sung Won Choi, Hafiz Muhammad Shahbaz, Jeong Un Kim, Da-Hyun Kim, Sohee Yoon, Se Ho, Jeong, Jiyong Park, and Dong-Un Leedeals with the simultaneous degradation of pesticides and microorganisms by Photolysis and TiO2 combined with UVC/VUV irradiation. Detailed analysis of degradation efficiency and kinetic parameters of AOPs for water contaminants and as well as for liquid and solid food materials can provide useful and interesting information for the science and practice, as well.
Manuscript is generally well written. Introduction provide a good summary of AOPs. But it can be noticed that Introduction does not provide a short summary of the environmental and health problems of pesticides. And the advantages/disadvantages of different AOPs are not discussed.
In materials and methods section 2.1 and 2.2.2, 2.3, 2.4-2.6 are given in detail, but pesticide concentration used for experiments ( in section 2.2) need explanation.
In section 3 experimental results are discussed in details. The results seemed to be realistic. But I recommend the authors to provide briefly the practical applicability of the different methods.
Comments, qustions:
C#1: In some cases the Abstract is too general ( ’…was little UV-TiO2 photocatalytic effect..’; ’.. slightly higher inactivation effects..’). I suggest to reconsider the phrasing of Abstract.
C#2: I suggest the authors to amend the Introduction section with a brief summary on the environmental problems of pesticides.
C#3: I suggest the authors to give a brief summary of advantage/disadvantage of different AOPs in Introduction section of MS.
C#4: Pesticide concentration range applied in experiments needs explanation (based on measurement, or references, or defined according to the experimental setups and analytic?)
C#5: I recommend the authors to provide briefly the practical applicability of the different methods (for example: matrix effect in a multicomponent system, such as surface/ground water) in Results and discussion section.
Q#1: During aerated VUV-TiO2 experiments how was possible to maintain the homogeneity of TiO2?
Q2: In combined UV and combined UV-TiO2 treatment of food system how changed the quality parameters (chemical, physicochemical/texture parameters, color, organoleptic characteristic) of carrot? Can be considered these treatments suitable to produce ’edible’ food product?
Author Response
Manuscript applsci-851060 by the authors of Sung Won Choi, Hafiz Muhammad Shahbaz, Jeong Un Kim, Da-Hyun Kim, Sohee Yoon, Se Ho, Jeong, Jiyong Park, and Dong-Un Leedeals with the simultaneous degradation of pesticides and microorganisms by Photolysis and TiO2 combined with UVC/VUV irradiation. Detailed analysis of degradation efficiency and kinetic parameters of AOPs for water contaminants and as well as for liquid and solid food materials can provide useful and interesting information for the science and practice, as well.
Manuscript is generally well written. Introduction provide a good summary of AOPs. But it can be noticed that Introduction does not provide a short summary of the environmental and health problems of pesticides. And the advantages/disadvantages of different AOPs are not discussed.
In materials and methods section 2.1 and 2.2.2, 2.3, 2.4-2.6 are given in detail, but pesticide concentration used for experiments (in section 2.2) need explanation.
In section 3 experimental results are discussed in details. The results seemed to be realistic. But I recommend the authors to provide briefly the practical applicability of the different methods.
è Thank you very much.
Comments, questions:
C#1: In some cases the Abstract is too general (‘…was little UV-TiO2 photocatalytic effect..’; ’.. slightly higher inactivation effects..’). I suggest to reconsider the phrasing of Abstract.
è We have specified the phrasing of Abstract in the revised manuscript (line 19-21, 24-25, 27-29).
C#2: I suggest the authors to amend the Introduction section with a brief summary on the environmental problems of pesticides.
è Pesticides do not always stay in the location where they were applied. They move along in the environment and often contaminate water, air and soil, even in remote areas. The toxicities of pesticides to organisms, including beneficial insects and non-target plants, can change the natural balance of the ecosystem by altering the environment to favor the pest [Braz. J. Microbiol. 2016, 47, 99-105]. This information is now included in the introduction section (line 39-43).
C#3: I suggest the authors to give a brief summary of advantage/disadvantage of different AOPs in Introduction section of MS.
è AOPs refer to oxidation processes that involve generation of hydroxyl radicals (OH·) and strong oxidants at sufficient concentrations. Hydroxyl radical (OH·) can be produced through the application including oxidizing agents (such as H2O2 and O3), UV irradiation, ultrasound (microbubble), electric current, and catalysts (such as Fe2+) [Curr. Pollut. Rep. 2015, 1, 167-176; Water 2020, 12, 102]. Hydroxyl radicals (OH·) are reactive oxidizing compounds with an oxidation potential of 2.80 eV, which is more reactive than both ozone (2.08 eV) and chlorine dioxide (1.36 eV). Furthermore, hydroxyl radicals (OH·) have advantages of no additional waste generation, no chemical toxicity, and no corrosiveness to equipment. However, the disadvantage of AOPs lies in their high cost due to the use of expensive reagents (e.g., H2O2), energy consumption (e.g., generation of O3), and unsatisfactory degradation efficiency (e.g., ultrasound alone) [Water 2020, 12, 102]. This information is now included in the introduction section (line 63-72).
C#4: Pesticide concentration range applied in experiments needs explanation (based on measurement, or references, or defined according to the experimental setups and analytic?)
è The pesticide concentration range applied in this study was set up taking into account that the MRLs of small fruits and vegetables are in the range of approximately 1 to 20 ppm. According to CODEX Alimentarius International Food Standards [http://www.fao.org/fao-who-codexalimentarius/codex-texts/dbs/pestres/pesticides/en/], MRLs for pyraclostrobin, boscalid, fludioxonil, and azoxystrobin were in the range of 0.01-40 ppm, 0.02-60 ppm, 0.01-60 ppm, 0.01-300 ppm, respectively. We have clarified this information in the revised manuscript (line 118-119).
C#5: I recommend the authors to provide briefly the practical applicability of the different methods (for example: matrix effect in a multicomponent system, such as surface/ground water) in Results and discussion section.
è In this study, degradation rate of pesticides slowed down along the course of the reaction time, attributed to the accumulation of oxidation intermediates. The decontamination technique for solid food materials is expected to be improved to achieve higher degradation rate by applying an open system with a continuous water flow through which clean water is supplied and treated wash water containing oxidizing intermediates is discharged. Meanwhile, in order to apply this decontamination technique to the liquid medium of a multi-component system such as surface/ground water, it is necessary to first understand the presence and mutual effects of components that reduce the reaction efficiency of the entire system, such as azoxystrobin in this study. Based on this information, decontamination techniques could be applied to multi-component systems in combination with the physicochemical methods and mechanical means including high turbulence, positive irradiation structures, static mixers or bubbles. This information is now included in the introduction section (line 380-389).
Q#1: During aerated VUV-TiO2 experiments how was possible to maintain the homogeneity of TiO2?
è Magnetic stirring at 300 rpm was applied during both mild and extra aeration under VUV-TiO2, too. The following treatments were carried out for aeration experiments: (i) magnetic stirring only (no aeration), (ii) air supply in a rate of 1.8 L/min with magnetic stirring (mild aeration), (iii) air supply in a rate of 20 L/min with magnetic stirring (extra aeration). We have clarified this information in the revised manuscript (line 143-146, 327-328).
Q#2: In combined UV and combined UV-TiO2 treatment of food system how changed the quality parameters (chemical, physicochemical/texture parameters, color, organoleptic characteristic) of carrot? Can be considered these treatments suitable to produce ’edible’ food product?
è We evaluated the effects of photolysis and UV-TiO2 photocatalytic treatments on the quality parameters of carrot samples by measuring TPA (texture profiles analysis), cutting forces, color (CIELAB value), and scanning electron microscopy. In briefly, there were no significant differences between control (bubble water wash) and the treated groups (combined UV and combined UV-TiO2) in all parameters. Although further study is needed to evaluate the changes in nutritional components and organoleptic quality of carrot, lycopene and total phenolic contents and antioxidant activities were not significantly changed in tomatoes after UV and UV-TiO2 treatment in our previous study. Of course, we aim to produce safe and ‘edible’ fresh produces using these treatments. We are preparing to publish a separate paper on the effects of photolysis and photocatalytic treatments in terms of physicochemical characteristics.
